# Social preferences in the public goods game–An Agent-Based simulation with EconSim

**Christoph Bühren** [1]*, **Jan Haarde**[2], **Christian Hirschmann**[2], **Janis Kesten-Kühne**[2]

**1** Faculty of Sport Science, Ruhr-University Bochum, Bochum, Germany, **2** Department of Economics, Clausthal University of Technology, Clausthal-Zellerfeld, Germany

* christoph.buehren@rub.de

## Abstract

Using a reinforcement-learning algorithm, we model an agent-based simulation of a public goods game with endogenous punishment institutions. We propose an outcome-based model of social preferences that determines the agent's utility, contribution, and voting behavior during the learning procedure. Comparing our simulation to experimental evidence, we find that the model can replicate human behavior and we can explain the underlying motives of this behavior. We argue that our approach can be generalized to more complex simulations of human behavior.

game–An Agent-Based simulation with EconSim. PLoS ONE 18(3): e0282112. https://doi.org/10.1371/journal.pone.0282112

**Data Availability Statement:** All data files for the agent-based model in EconSim are available from the open science framework: (https://osf.io/xhgfq).

## 1. Introduction

Economic theories assume (over)simplified models without complex interaction between economic actors and markets. The partial equilibrium model, for example, focuses on single markets without interdependencies between markets. Laboratory experiments can be used to test economic theories with humans. Experiments are a tool to examine a broad spectrum of economic problems, such as human behavior in competitive or cooperative games. The public goods game models dilemmas that modern society faces every day–from the cooperation of roommates or team members to the cooperation of the entire world in combating climate change [1]. In experiments, subjects decide in simplistic and unnatural environments. Hence, it seems doubtful that it will be possible to derive useful insights into the complexity of real economic behavior.

Agent-based simulations, on the other hand, can model this complexity. [2] set the foundation for agent-based modeling in the social sciences by showing that agents with "zero intelligence" can be sufficient to ensure a reasonably high allocative efficiency on markets [3]. However, as these models substitute human subjects with robot agents, realistic behavior has to be implemented to yield credible results. Experiments and agent-based models have already been successfully combined. [4] studied price expectations simultaneously with human and robotic subjects. Also, the results of laboratory experiments can be used to calibrate realistically behaving agents [5].

**Funding:** The authors received no specific funding for this work.

**Competing interests:** The authors have declared that no competing interests exist.

In theoretical economic models, the neoclassical homo oeconomicus is the standard assumption for the analysis of games and the determination of equilibrium strategies. By default, the homo oeconomicus is defined as a self-interest-maximizing individual implicating non-cooperative behavior in social dilemmas. However, this theoretical construct does not explain what is usually observed in experiments. In contrast to pure rational choice, at least in the first rounds of the public goods game, participants cooperate [6]. Behavioral models of social preferences better describe the observed behavior. In these models, the outcomes of others influence the own evaluation of an allocation. However, assumptions that do not solely rely on self-interest do not necessarily contradict neoclassic theory [7]. The Fehr-Schmidt model [8] allows people to be self-interested and inequity averse. Furthermore, it distinguishes between advantageous and disadvantageous inequity. Bolton and Ockenfels's model of inequity aversion [9] does not differentiate between these two forms. Moreover, [10] consider the efficiency of the distribution and minimal outcome in a group.

While some people even cooperate in social dilemmas, such as conditional cooperators [11] or altruists [12] a lot of people tend to be selfish. Consequently, it is important to study mechanisms that may increase cooperation levels in groups. A powerful tool to enforce cooperation in public goods games is the punishment of free-riders. Experimental economists introduced many different punishment institutions. Informal punishment means that individuals may choose to sanction other group members directly [13]. On the other hand, formal punishment imposes a sanctioning rule, which specifies the occurrence and the extent of punishment [14, 15]. [16] compared both schemes in a setting with endogenous punishment, i.e., subjects voted either for formal or informal punishment. Evolutionary game theorists studied the effects of punishing individuals in the public goods game with replicator dynamics (see [17], for a recent overview). For instance, [18] show that tax-based pure punishment (not helping others but punishing free-riders) has an evolutionary advantage in sustaining cooperation over pure punishment without tax. Moreover, [19] find that a formal sanctioning institution that combines reward and punishment promotes cooperation at a lower cost compared to either reward or punishment alone.

The experimental results of [20]'s public goods game with controlled group formation and endogenous formal punishment indicate heterogeneous behavior of different subject groups in similar situations. While groups of "low cooperators" need punishment institutions to enforce cooperation, "high cooperators" act in a socially beneficial way even without punishment. Social preferences may explain these observations. Whereas subjects can be assumed to follow their own social preferences when deciding how much to contribute to the public good, we can directly incorporate the expected preference structures in agent-based simulations. Thus, possible motives of human behavior in dilemma situations could be better explained by comparing the experimental and simulation results [21].

In our paper, we show that reinforcement learning can replicate human behavior. For that reason, we construct a model of social preferences and use it in an agent-based simulation of a public goods game with a setting similar to the design of [20]. Our model consists of different motives that may explain the behavior observed in the experiment. By weighting these motives, we create utility functions embodying different preference structures, incorporated into our agents. With reinforcement learning, homogenous agents learn to play their optimal strategies in three punishment settings replicating the behavior of the human subjects in [20]. Reinforcement learning is a trial-and-error learning paradigm, based on the principle of operant conditioning, assuming learners evaluate actions by trying them and getting rewards as feedback. Operant conditioning is a psychological framework for learning based on the law of effect by [22], which states that actions, which reward the individual, are executed more often. Over time, the learner explores good actions for given situations and exploits the rewards of the

actions, when satisfactory reward levels are reached. Reinforcement learning might be the closest to human learning among the various machine learning algorithms as it is based on biological learning [23]. Thus, we use it to replicate human behavior in our agent-based model. The game-theoretic or simulative reproduction of human behavior is not only interesting in theoretical economic models but also in practical applications such as wireless network sharing [24], traffic simulation [25], the simulation of escape routes in case of emergencies [26], or delay management in railway networks [27].

Further contributions of our paper are the replication of the public goods game of [20] in an agent-based simulation and the introduction of a newly combined social preference model. Additionally, we show that human behavior in public good experiments can be replicated using simple motives with only a few parameters. Suppose the experimental results constitute a Petri dish with some bacterial cultures flourishing in it. We do not know which motives were the exact drivers of the behavior. Instead, we reproduce several comparable Petri dishes with our model of social preferences and a reinforcement-learning algorithm as ingredients.

## 2. Effect of information on the demand for punishment in a public goods experiment

[20] combine endogenous punishment institutions in the public goods game ("institution formation game") with a sorting mechanism based on participants' previous contributions in a one-shot public goods game ("sorting game"). Thereby, homogeneous groups of participants with similar ex-ante cooperation levels play the public goods game, in which they can vote for punishment institutions. To examine the role of information on the demand for punishment, the authors implemented two treatments: *Sorted* and *Sorted-Info*. In both treatments, group members in the institution formation game are sorted–they contributed similar amounts of tokens in the sorting game. Thus, their initial level of cooperativeness is similar. In *Sorted*, players only learn about their group members' contributions during the institution formation game; in *Sorted-Info*, they are additionally informed about their group members' initial contributions in the sorting game.

### 2.1 Experimental design

The experiment of [20] consisted of two games–the sorting and the institution formation game. In the sorting game, subjects were randomly shuffled into groups of five to play a standard one-shot public goods game. In the game, the endowment was set to 350 tokens, and the participants simultaneously decided on their individual contribution $g_i \in [0,350]$ to the public good. The marginal product of the public good is $a = 1.5$ and the marginal per capita return $\frac{a}{n} = 0.3$. These parameters were chosen to guarantee the basic characteristics of the public goods game ($a > 1$ and $\frac{a}{n} < 1$) and appropriate payoffs for the participants. In this case, the payoff function can be written as

$$\pi_i = 350 - g_i + 0.3\sum\nolimits_{j=1}^{5} g_j. \tag{1}$$

The endowment in the sorting game was set at 350 tokens because the authors tried to induce enough variation in the sorting game contributions and a less obvious focal point in the middle of the scale–as compared to the more standard 100 tokens, in which initial contributions may be clustered around 50 (see, e.g., [28]). Based on the contributions in this sorting game, groups of like-minded cooperators were formed for the institution formation game. High cooperators contributed more than 250 tokens, low contributors less than 150 tokens, and middle cooperators between 150 and 250 tokens.

In the *Sorted-Info* treatment, participants received more information on their group members of the institution formation game than participants in the *Sorted* treatment. In the *Sorted-Info* (but not in the *Sorted*) treatment, participants knew the sorting game contributions of their group members in the second game. This second game consisted of six phases with four rounds each with an endowment of $E = 100$ tokens every round. In the institution formation game, the authors used this standard endowment to make their results better comparable to previous literature. After each round, the contributions of the group members are displayed in random order. At the beginning of every phase, the groups vote on a punishment institution. The payoff function for a given round in a phase without punishment is

$$\pi_i = 100 - g_i + 0.3\sum\nolimits_{j=1}^{5} g_j. \tag{2}$$

If three or more players voted to play with punishment, the group had to decide between mild and severe punishment. In every round, every group member had to pay an institutional fee of 5 for mild or 20 for severe punishment deducted after the contribution stage. Every person contributing less than the maximum of 100 tokens to the public good automatically had to pay a fine of 50 under mild punishment and 90 in the severe punishment scheme. With punishment, the payoffs are calculated by

$$\pi_i = 100 - g_i + 0.3\sum\nolimits_{j=1}^{5} g_j - f(k) - sgn(100 - g_i) \cdot p(k), \tag{3}$$

where $f(k)$ is the fee and $p(k)$ the fine in institution k. The signum function $sgn(\cdot)$ equals 1 when its argument is strictly positive (i.e., the contribution $g_i$ is less than 100) and 0 otherwise. Table 1 summarizes the punishment institutions.

Note that mild punishment does not solve the dilemma structure of the game, even though it decreases the difference in payoff between complete defection and complete cooperation. In contrast, the payoff maximizing strategy under severe punishment is to make a full contribution. These formal punishment institutions are similar to [14]. Even the mild punishment option has the potential to increase cooperative behavior because a fixed fine is imposed on players who contributed less than their whole endowment. However, the net effect of the institutions on payoffs is less clear because their implementation is costly.

## 2.2 Results

Groups of high cooperators achieved high contribution levels irrespective of the institution. Because of the costs of punishment, the average payoff is lower for cooperative groups with punishment than without. The contributions of middle and especially of low cooperators were significantly higher with punishment than without. However, middle cooperators only earned significantly more with punishment in *Sorted* and low cooperators in *Sorted-Info* (see Fig 2 in [20]).

As expected, in the information treatment, low cooperators had the highest demand for punishment. They implemented severe punishment significantly more often than high and middle cooperators (see Fig 5 in [20]). High cooperators had by far the lowest demand for punishment in *Sorted-Info*. However, without the information on the groups' initial level of

**Table 1. Punishment institutions.**

| Punishment institution | Fee $f(k)$ | Punishment $p(k)$ |
|---|---|---|
| No ($k = 1$) | 0 | 0 |
| Mild ($k = 2$) | 5 | 50 |
| Severe ($k = 3$) | 20 | 90 |

cooperation, the voting behavior throughout the institution formation game is detrimental to most of the groups' payoffs. This finding is remarkable because, also in *Sorted*, subjects know the contributions during the second game but do not seem to learn from it. Even when low cooperators implemented severe punishment, they did not behave wisely in this institution. The main reason is that some of the low cooperators are willing to sacrifice money as long as they earn more than others in their group.

## 3. An agent-based model of the public goods game in EconSim

This section introduces an agent-based simulation of the public goods game in EconSim, a framework to create complex models, to investigate if agents with social preferences and reinforcement learning produce similar results as human participants in the experiment. We propose a model of social preferences based on different forms discussed in the previous literature (see Section 1) and the possible motives of subjects derived from the experimental observations.

### 3.1 The modular framework EconSim

EconSim allows complex agent-based models based on predefined and highly adjustable components [29, 30]. It distinguishes between different types of agents: households, firms, states, and central banks. Households may represent consumers who focus on maximizing their utility, while firms represent the producers in an EconSim model. The state's role is to deliver the legal framework of the simulation environment, which can be endogenous through agents' voting decisions. Central banks may be used to control the supply of money, which can be used as a general numéraire.

Goods are pivotal for a simulation with EconSim. They are not only consumables or production units but also indicate a group affiliation. They can be traded on markets or transformed according to transformation plans that define how the good is produced, stored, recycled, transformed, or scrapped. The model can incorporate an arbitrary number of goods with different relations to other goods. For instance, two goods representing raw materials may transform into a consumable good that is consumed by households to gain utility. Trades are executed on markets that connect buyers and sellers using a set of market mediation rules. As another important part of the simulation, institutions can be implemented that influence the rules of the simulation environment, the behavior of agents, and especially the action space of state agents.

A public goods game could be implemented in agent-based simulations that are simpler. However, we chose EconSim because this framework provides powerful and ready-made algorithms, which are verified and validated thoroughly. It is easy to change the properties and the decision-making in this simulation software. Furthermore, the modular architecture of EconSim allows us to expand our model with additional elements. For instance, we could introduce a revolution mechanism into the model or extend the game to a more complex model, which endogenizes the agents' endowment. EconSim provides tools for future research to easily adapt our model, which is calibrated with experimental data.

### 3.2 Reinforcement learning

Traditional economic theory assumes rational decision-makers, which is challenged by experimental and empirical evidence. Reinforcement learning, however, applies bounded rationality by simulating non-optimizing but satisficing behavior similar to human decision-making [31]. It does not require specific information about the environment or other players. The agent chooses an action in a given state based on expected rewards. According to the action, the state of the simulation changes, and the reward function calculates the agent's factual reward. As a consequence, the agent learns by updating the expectations of the rewards [21, 32]. "Reward"

is the usual term for a measurement of success in the context of reinforcement learning. Later, we will use utility function values as rewards. Payoffs in the public goods game are not to be misunderstood as rewards because the utility function value may differ.

Agent-based models provide an environment, in which heterogeneous agents interact with each other [33]. Each period in an agent-based simulation can be viewed as one state of a set of possible states. Agents can choose an action of a set of possible actions, which leads to a reward at the end of a period. Reinforcement learning enables agents to optimize their expected reward over time. Similar to other machine learning techniques, the modeler does not have to implement pre-determined rules of behavior, which increases the realism of the simulation. By varying the reward function and/or the learning parameters, different types of agent behavior can be created. Reinforcement learning allows for adaptive behavior and learning on the individual level. Therefore, its use in an agent-based model to simulate decision-making and learning is convenient. According to the review of [34], reinforcement learning is the most prevalent machine learning method in agent-based modeling. Either single agents learn to maximize their own goal, or multiple agents learn to adapt towards a common target [35].

In our agent-based model, we use a simplified version of Roth and Erev's reinforcement learning algorithm [36]–a variant of the normalized exponentially smoothed attractivity-based strategy selection algorithm (NESASS) by [30]. Every possible action possesses an attractivity value, which determines the probability of choosing this action and is initialized either randomly or by a pre-defined value. The following function defines updating of the attractivity $q$ $(a, t)$ at the end of period t and, thus, the learning process:

$$q(a, t) = \begin{cases} \alpha^{new} \cdot r(s_t, a) + \alpha^{old} \cdot q(a, t-1), & \text{if } a = a_t \\ (1 - \varphi) \cdot q(a, t-1), & \text{if } a \neq a_t \end{cases} \quad (4)$$

The new attractivity value $q(a, t)$ for the chosen action $a_t$ is a weighted sum of the old value $q(a, t-1)$ and the received reward $r(s_t, a)$ of the action. In this algorithm, agents do not have information about the current state $s_t$. The parameters $\alpha^{new}$ and $\alpha^{old}$ are the weighting factors. If $\alpha^{new} + \alpha^{old} = 1$, this update procedure follows a classic exponential smoothing. Not chosen actions become devalued in attractivity controlled by a depreciation parameter $\varphi$. Note that reward and attractivity are different. Alternatively, the attractivity value of an action can be interpreted as the agents' reward expectation for this particular action. Rewards are the metric for the experienced success in a particular period.

Based on the attractivity values, the probability $p(a, t)$ that action $a$ is chosen at any time $t$ is defined by the following softmax function:

$$p(a, t) = \frac{\exp\left(\frac{q(a,t)}{q^{max} \cdot \mu(t)}\right)}{\sum_{\tilde{a} \in A} \exp\left(\frac{q(\tilde{a},t)}{q^{max} \cdot \mu(t)}\right)}. \quad (5)$$

The variable $q^{max}$ is a hyper-parameter and denotes the attractivity value of the currently most attractive action, and $A$ is the set of actions available. To dynamically adapt the learning process, the temperature parameter $\mu$ regulates the sensitivity for choosing the best action regarding its attractivity. With low values of $\mu$, the agent tends to greedily choose the best action, even when it is only slightly better. The limit of $p$ is

$$\lim_{\mu \to \inf} p(a, t) = \frac{1}{|A|} \quad (6)$$

and equivalent to drawing from a discrete uniform distribution. Thus, high values of $\mu$ indicate

more explorative behavior. The $\mu$-value should depend on the learning process and should consider uncertainty. In our model, we update $\mu(t)$ as follows:

$$\mu(t) = \begin{cases} max\left(\underline{\mu}, \ \mu(t-1) \cdot (1 - \mu_\Delta)\right), & \text{if } a_t^* = a_{t-1}^* \\ min(\mu(t-1) \cdot (1 + \mu_\Delta), \bar{\mu}), & \text{if } a_t^* \neq a_{t-1}^* \end{cases} \quad (7)$$

This update scheme is active if, in the last two periods $t$ and $t-1$, there is exactly one action with the highest attractivity value ($a_t^*$ or $a_{t-1}^*$), otherwise $\mu(t) = \mu(t-1)$. Therefore, it can be active at the end of period 2 for the first time when at least one updated attractivity is higher than the initial value. For instance, if we initialize the attractivity vector with 50 and the chosen actions achieve a reward of 40 and 45, then the update scheme is not active. This is because there are multiple actions with the same (depreciated) attractivity value of $50 \cdot (1 - \varphi)^2$, assuming a low depreciation parameter. In the case of $a_t^* = a_{t-1}^*$, $\mu$ decreases because no new best action is found in period $t$. If this happens regularly, the action space seems to be sufficiently explored. If not, there is uncertainty about the optimum within the action space, and $\mu$ rises. Additionally, we introduce the parameter $\varepsilon$ (epsilon greedy). The product $\varepsilon \cdot \mu(t)$ delivers a small probability with which the agent chooses from a discrete uniform distribution, ignoring current attractivity values. In a dynamic environment, where best actions may change over time, $\varepsilon$ induces the exploration of new actions even if the optimum seems to be found. As we want to achieve a stable convergence without random decisions, we multiply $\varepsilon$ by $\mu(t)$ to reduce explorative behavior after agents' learning is achieved (low $\mu(t)$). The $\varepsilon$-greedy parameter is not to be misunderstood as a control parameter for agents to act greedy. It is a means to manage the exploration and exploitation problem in a reinforcement learning approach. With no or too low $\varepsilon$-greedy, learning might converge too fast into a local optimum as the strategy space is not explored well enough. If the $\varepsilon$-greedy value is too high, we would expect no convergence at all as random decisions dominate. See [23] for a detailed description. We provide a pseudocode of our algorithm in S1 File.

With the described reinforcement-learning algorithm, we present a way how agents learn to choose their cooperation level according to the resulting rewards (payoffs) of their different decisions. In Section 3.6, we apply this learning mechanism to the voting on the punishment institutions. Reinforcement learning is also known as operant conditioning and describes a trial-and-error process until a sufficiently good decision in a given situation is discovered. This process of exploitation and exploration is regularly found in animals and humans [37]. Thus, the algorithm used can mimic human behavior.

## 3.3 The setup in EconSim

S1 Fig shows the graph of our agent-based model. It replicates the experimental design of [20] by using households as agents, a public good, and punishment institutions. We use three types of agents to resemble the outcome of the experiment's sorting mechanisms. The model consists of a standard good serving as tokens endowed by the households. At the beginning of each period, the households decide how much to contribute to the public good. Based on the contributions, payoffs and objective function values are calculated. Moreover, the simple majority of a group of five households decides if a punishment institution is implemented and (if yes) whether this institution is mild or severe. Fig 1 shows a flow chart of the essential modules of our simulation model.

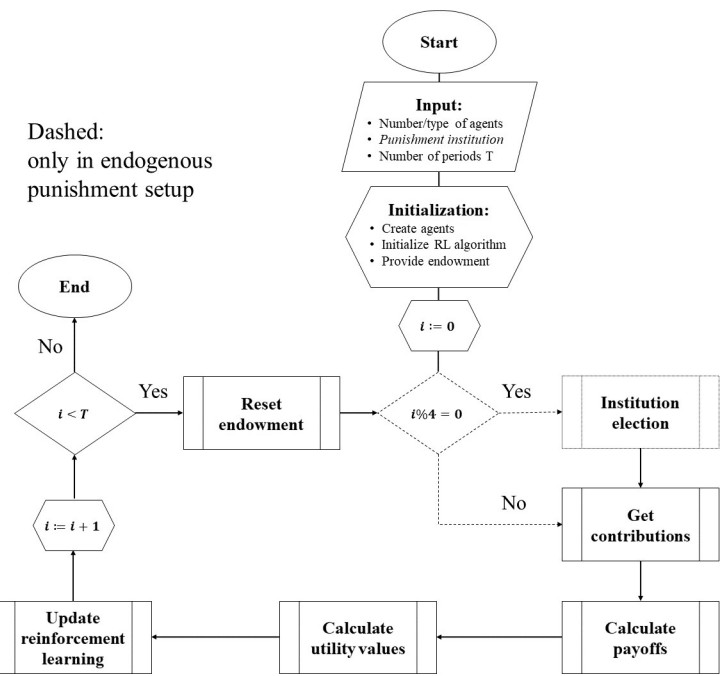

**Fig 1. Flow chart of essential modules in our simulation model.**

## 3.4 Our model of social preferences

The goal of our agent-based simulation is to show that a simple learning algorithm combined with a model of social preferences can replicate the experimental findings of [20]. We developed this model of social preferences inspired by possible motives that may explain subjects' behavior in the experiment. It describes an agent by a weighted average of five different motives, defined by the utility functions presented in Table 2. A purely selfish agent only looks at its own payoffs. The utility of an inequity-averse agent is high if all agents have relatively equal payoffs. Competitive agents want to earn more than others [38]. The fourth motive aims to maximize social welfare, and the utility of altruistic agents simply increases with their contributions. We normalize the utilities to a range from 0 to 100 to make them better comparable across motives and types of agents.

In line with other models of social preferences, agents may combine different motives. The model of [9] combines selfishness as well as advantageous and disadvantageous inequity

**Table 2. Formal definitions of social preferences and their marginal utility.**

| Social preference | Formal definition | Marginal utility $\frac{\partial u_{ij}}{\partial g_i}$ |
|---|---|---|
| 1) Selfishness | $u_{i1} = \dfrac{100 - g_i + 0.3 \sum_{j=1}^{n} g_j - sgn(100 - g_i) \cdot p(k)}{1.5}$ | $-\dfrac{0.7}{1.5}$ |
| 2) Inequity aversion | $u_{i2} = 100 - \dfrac{\sum_{j=1}^{n} (g_j - \bar{g})^2}{25 \cdot n}$ | Dependent on difference between own and others' contribution |
| 3) Competitiveness | $u_{i3} = 50 + \dfrac{\sum_{j=1}^{n} (g_j - g_i)}{2 \cdot (n-1)}$ | $-\dfrac{1}{2}$ |
| 4) Social welfare | $u_{i4} = \dfrac{\sum_{j=1}^{n} g_j}{n}$ | $+\dfrac{1}{5}$ |
| 5) Altruism | $u_{i5} = g_i$ | $+1$ |

aversion. [5] combine selfishness with social welfare and reciprocity. To keep it simple, we assume an additive combination of different motives, where $w_{ip}$ is the weight of a social preference $p$ in agent $i$. Regardless of the motives, the fee $f(k)$ causes every agent to have a clear tendency to avoid punishment when it is not necessary (see Section 2.1).

$$U_i = \sum_{p=1}^{5} w_{ip} \cdot u_{ip} - f(k) \tag{8}$$

This objective function will serve as the reward function. Thus, it is a measure of an agent's success in single rounds. The reinforcement-learning algorithm is used to make agents decide and learn autonomously based on success. In our simulation, agents decide about their contribution level in each round. The attractivity of a contribution is its expected utility function value in the chosen punishment institution. This attractivity is updated in every round. The new expected utility of the chosen contribution level is the weighted sum of the old and the new value. All other contributions' expected utility values are depreciated by one percent in this round. Thus, our agents remember old experiences but also forget about the success of contribution levels that were not played for a while. Moreover, after round 10, agents decide with a 20 percent probability by imitating the previous round's decision of another agent. With the resulting objective function value, the attractivity of the imitated decision is updated. Imitation of others is also a form of exploration. Agents may learn from other agents' contributions by imitating a random co-player. In some cases, it will not be helpful to only learn from the actions that lead to the highest reward for another agent. Examples are dynamic and unstable environments or agents' heterogeneous preference structures. In our simulation, when severe punishment is active, an agent with a contribution of less than 10 will have a higher payoff than agents contributing the efficient amount of 100. Only learning from the most successful agent would be inefficient in this case.

## 3.5 Exogenous punishment institutions

The agents in our simulation represent the three types described in [20]: low, middle, and high cooperators. Using their objective function, agents can be modeled by choosing the weights $w_{ip}$ for every given social preference motive (see Tables 2 and 3). To predict the agents' tendency for cooperation, we calculate their marginal utility of an additional unit of contribution for a given weight combination in Table 3. In this analysis, we neglect the effect of inequity aversion because we assume homogenous agents. For them, inequity aversion will not change the game-theoretic prediction because all agents will maximize their utility by the same contribution level. However, inequity aversion affects our simulated results. See Section 3.7 for a detailed description. For the social preference motives, we chose the weights of Table 3 such that low cooperators have a negative marginal utility and high cooperators have a positive marginal utility. In contrast, the middle cooperator has no clear preference for or against cooperative behavior.

**Table 3. Definition of agent types.**

| Agent | $w_{i1}$ | $w_{i2}$ | $w_{i3}$ | $w_{i4}$ | $w_{i5}$ | Marginal utility $\frac{\partial U_i}{\partial g_i}$ |
|---|---|---|---|---|---|---|
| Rational low cooperator ($i = 1$) | 1 | 0 | 0 | 0 | 0 | -0.46 |
| High cooperator ($i = 2$) | 0.45 | 0.1 | 0 | 0.2 | 0.25 | 0.08 |
| Middle cooperator ($i = 3$) | 0.55 | 0.1 | 0 | 0.15 | 0.2 | -0.03 |
| Competitive low cooperator ($i = 4$) | 0.6 | 0 | 0.4 | 0 | 0 | -0.41 |

**Table 4. Parameter settings of the reinforcement learning algorithm.**

| Reinforcement learning parameter settings | | | |
|---|---|---|---|
| Epsilon greedy | $\varepsilon = 0.01$ | Initial temperature | $\mu = 1$ |
| Initial attractivity | $q(a, 1) = 50$ $\forall a \in A$ | Temperature variation | $\mu_{var} = 0.05$ |
| Exponential smoothing new value | $\alpha^{new} = 1$ | Lower/upper bound temperature | $\left[\underline{\mu}, \bar{\mu}\right] = [0.01, 1]$ |
| Exponential smoothing old value | $\alpha^{old} = 0$ | Attractivity depreciation | $\varphi = 0.005$ |
| Imitation probability | $p_{imit} = 0.2$ | Imitation of... | Any agent |

During our simulation, weights are kept constant. The main setup follows the experimental design of the institution formation game by [20]. Instead of 24 rounds, however, our simulation runs 300 periods to allow agents' learning based on reinforcement learning. Table 4 summarizes the parameter settings for the reinforcement-learning algorithm. As social preferences are directly modeled into the agents, the sorting game of [20] is not needed. For the simulations, we build different types of agents, based on our model of social preferences, who may behave similarly to the subjects of the experiment. Similar to [20], we create groups of five agents following the same type of cooperation. Our simulation resembles the *Sorted* treatment of [20]–agents are sorted into groups of "like-minded people" but are not informed about any sorting criteria. In contrast to their design, we first exogenously impose one of the three punishment institutions before introducing endogenous institutions (see Section 3.6). Exogenous punishment institutions are standard in public goods experiments (see [39], for a meta-analysis), endogenous punishment institutions, in which subjects vote on the institution, are relatively new [16, 40]. The datasets of our simulation results are published at https://osf.io/xhgfq, and the results of the different types of agents are described below.

**Rational low cooperators.** Our simulation starts with solely self-interested agents (type $i = 1$, rational low cooperators) whose only interest is to maximize their own payoff without looking at the results of others. We use this first step to validate the program by comparing the results to game-theoretic predictions. Furthermore, we compare the results to the low cooperators of [20]. We compare the average contributions and payoffs under the different institutions in the experiment with our simulation: a single run and the arithmetic mean of 100 simulation runs (Figs 2 and S3 and Table 5). Learning is highly dependent on coincidence as agents decide randomly at the beginning and have to find their individual optima. Furthermore, agents might be tricked by the dynamics of the simulation. If an agent plays its personal optimum, but all others randomly contribute at a relatively low level, this agent will undervalue this choice. Subjects in the experiment do not have an underlying objective function or at least it is not directly observable, but the patterns emerging from their behavior may be reconstructed by using only simple motives like the ones described above.

S3 Fig shows the contributions and payoffs of a group of five rational low cooperators, while Fig 2 shows the average contributions and payoffs after 100 simulation runs. In the beginning, agents decide randomly. Yet they quickly learn that lower contributions lead to higher individual payoffs in the setting without punishment. This behavior, however, leads to low payoffs of around 100 on average for every group member, which corresponds to the Nash equilibrium. Similarly, in the experiment, low contributors in *Sorted* earned an average payoff of 108 without punishment.

Under mild punishment, we observe similar results as in the simulation without punishment. This is because the Nash Equilibrium does not change with mild punishment–it is designed as a non-deterrent sanction scheme [14]. Agents still maximize their individual

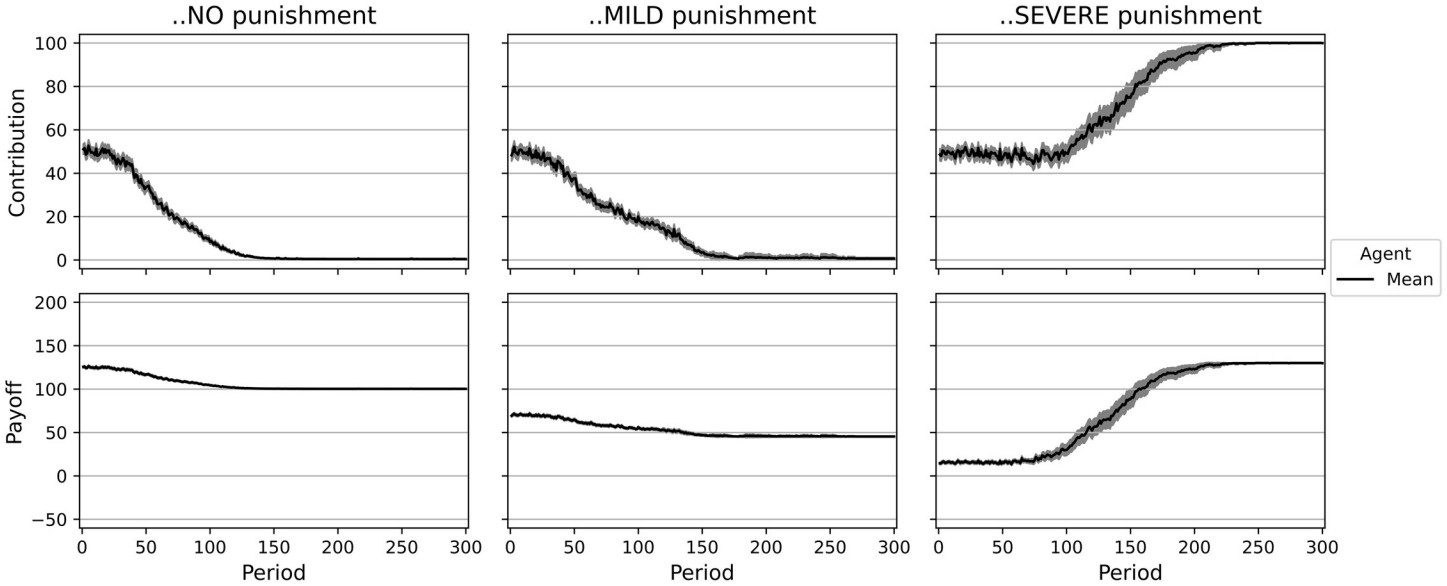

**Fig 2. Dynamics of contributions and payoffs of low cooperators under no, mild, and severe punishment.** Arithmetic mean of 100 simulation runs. In gray: confidence interval (95%) around the mean.

payoff by free-riding. Due to the costs of punishment (5 tokens fee and 50 tokens fine), the agents' average payoff is reduced to 45.31 tokens in later periods using this institution. Similarly, low cooperators in the *Sorted* treatment of [20] earn less with punishment than without–but not significantly less and, with an average of 101 tokens in the mild institution, much more than agents in the simulation.

The individual optimum with severe punishment is identical to the social optimum, namely, every agent fully contributes to the public good. Even though some agents seem to

**Table 5. Contributions and payoffs under exogenous punishment in the last 20 periods of 100 simulation runs.**

| Agent type | Punishment institution | Average contribution | Average payoff |
|---|---|---|---|
| Rational low cooperator ($i = 1$) | No | 0.45 (0.78) | 100.22 (0.39) |
| | Mild | 0.62 (2.72) | 45.31 (1.36) |
| | Severe | 100.00 (0.13) | 129.99 (0.13) |
| High cooperator ($i = 2$) | No | 93.92 (5.11) | 146.96 (2.56) |
| | Mild | 98.20 (3.76) | 131.60 (23.23) |
| | Severe | 99.99 (0.44) | 129.99 (0.62) |
| Middle cooperator ($i = 3$) | No | 56.02 (24.13) | 128.01 (12.07) |
| | Mild | 93.12 (16.83) | 131.55 (27.32) |
| | Severe | 100.00 (0.00) | 130.00 (0.00) |
| Competitive low cooperator ($i = 4$) | No | 0.34 (0.68) | 100.17 (0.34) |
| | Mild | 0.20 (0.57) | 45.10 (0.29) |
| | Severe | 2.40 (2.40) | -7.00 (19.58) |

In brackets: standard deviations.

have problems finding the optimal contribution level, imitation leads to the optimum with a higher average payoff than without punishment. Note that imitation will not necessarily help if agents learn from the best agent only. Agents with a lower contribution than ten tokens despite the punishment get a higher payoff than fully contributing ones. Therefore, only these agents would be imitated, resulting in a worse payoff for the imitating individual. This is why in the case of imitation, the last decision of a random agent is copied. In the end, agents earn an average payoff of 129.99 (reduced by the fee of 20) and commit to complete cooperation. In the experiment, low cooperators in *Sorted* contributed only 84 out of 100 under the severe punishment institution. They even earned slightly less than under the no punishment institution (payoffs of 106 vs. 108). The type rational low cooperator behaves as a homo oeconomicus after learning is converged. Thus, our model can match classical economic theory.

**High cooperators.** Next, we try to mimic the behavior of the high cooperators in the experiment of [20]. In the simulation, this type of agent ($i = 2$) uses an objective function that consists of selfishness and altruism, social welfare, and some inequity aversion (see Table 3). This agent type resembles subjects who are interested in their own payoff. However, altruistic preferences, concerns for social welfare, and inequity aversion dominate the agents' utility function of type 2. S4 Fig shows the development of contributions and payoffs of five high cooperators in a single run, while Fig 3 represents the average results of 100 simulation runs with this setting. Even though agents do not necessarily learn to contribute the entire endowment, there is a clear tendency to choose a contribution level slightly below full cooperation leading to an average payoff of 146.96 without punishment. Likewise, in the experiment, it was 141 tokens. Under mild punishment, the high cooperator agents increase their contribution to almost full cooperation with 98.20 tokens. However, this small increase does not compensate for the costs of punishment leading to a slight decrease in payoff, in line with the experimental

## High cooperators
## (mean of 100 runs)

### under..

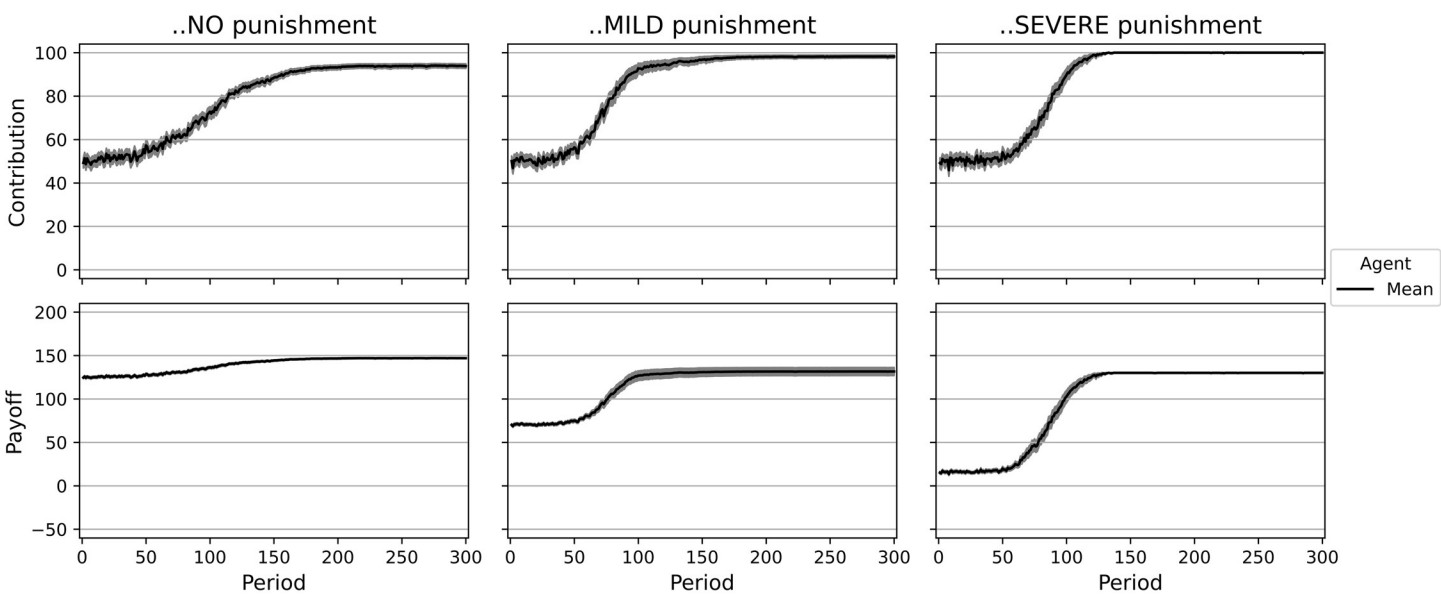

**Fig 3. Dynamics of contributions and payoffs of high cooperators under no, mild, and severe punishment.** Arithmetic mean of 100 simulation runs. In gray: confidence interval (95%) around the mean.

findings. Severe punishment makes it even worse and results in a payoff of around 130 for high cooperators in the simulation and 122 in the experiment.

**Middle cooperators.**   Between the types of low and high cooperators, we calibrated the type of middle cooperator (see Table 3). This type of agent is more self-interested, less altruistic, and less inequity-averse than the high cooperator. Yet agents of type 3 are less self-interested than rational low cooperators. Figs 4 and S5 show the simulation results for these agents in a single simulation run and, on average, after 100 runs. After the learning periods, their average contribution is around half of the endowment under no punishment. However, middle cooperators seem to agree less on a cooperation level than the other types. Thus, their payoffs are heterogeneous with an average of 128.01 tokens. In the experiment, it was only 114 tokens. Under mild and severe punishment, the agents of type medium cooperator (almost) fully cooperate. In the experiment, punishment also leads to significantly higher payoffs for middle cooperators in *Sorted*, but not to full cooperation.

**Competitive low cooperators.**   While the middle and high cooperators in the simulation reflect the behavior observed in the experiment quite well, agents of type $i = 1$ cannot explain the suboptimal contributions of low cooperators in *Sorted*. Indeed, this was one of the most challenging results of [20]. Some participants seem to be willing to earn more than others in their group regardless of the total amount of the payoffs. This behavior inspired our competitive type of agent ($i = 4$). These agents are still mainly self-interested, similar to the type of low cooperator. However, they are additionally described by competitive traits and modest social welfare preferences. Figs 5 and S6 show that the competitive low cooperators free-ride under any institution at the end of the learning process. Under no and mild punishment, the results are similar to the rational low cooperator. Yet under severe punishment, after a longer learning

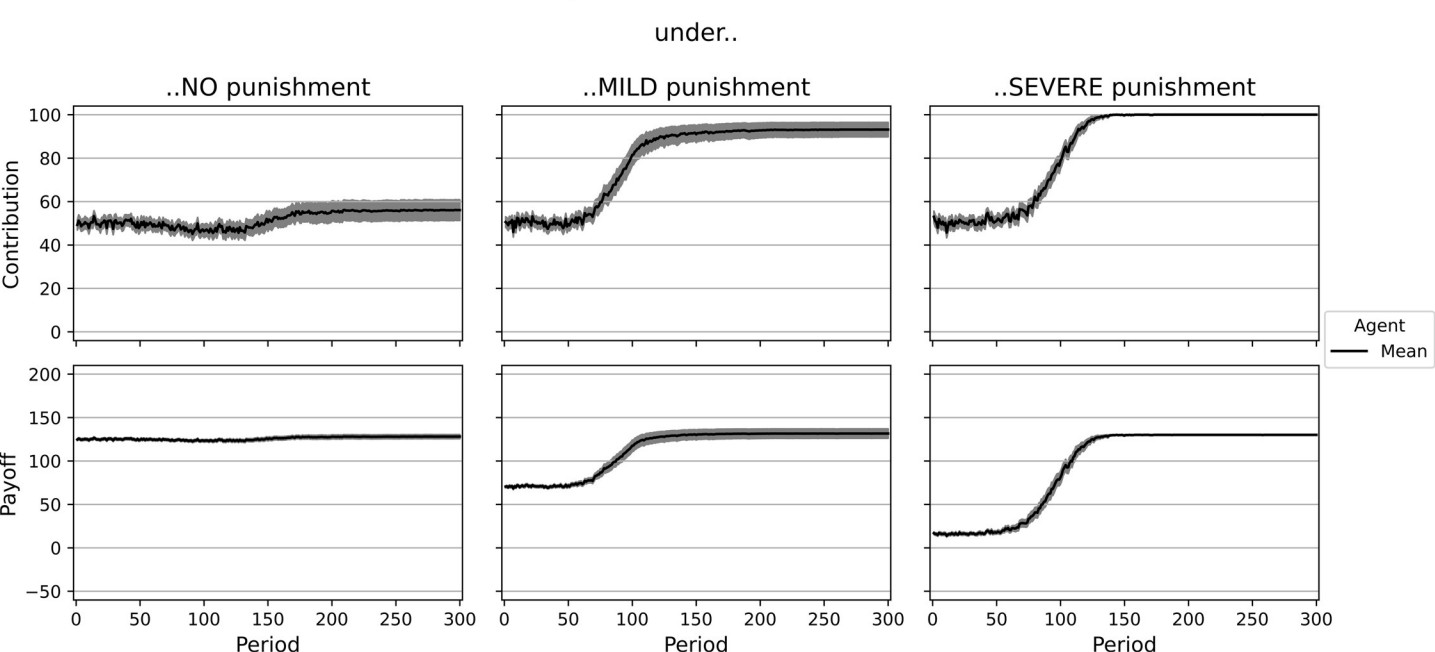

**Fig 4. Dynamics of contributions and payoffs of middle cooperators under no, mild, and severe punishment.** Arithmetic mean of 100 simulation runs. In gray: confidence interval (95%) around the mean.

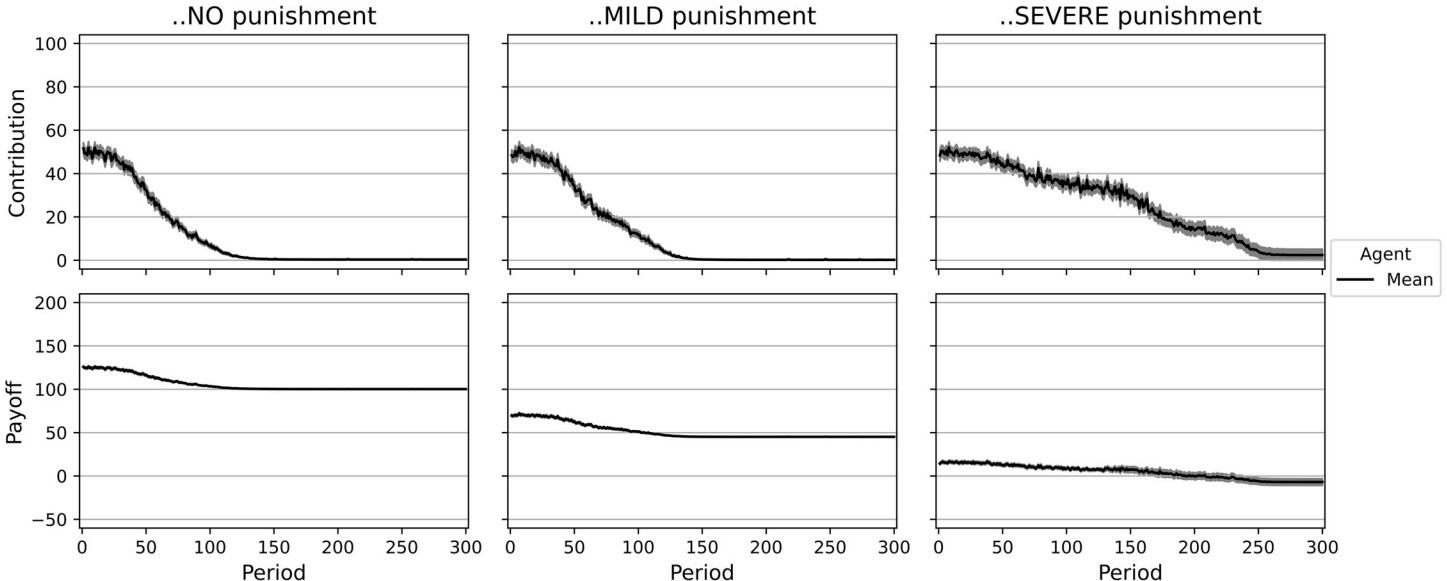

**Fig 5. Dynamics of contributions and payoffs of competitive low cooperators under no, mild, and severe punishment.** Arithmetic mean of 100 simulation runs. In gray: confidence interval (95%) around the mean.

process, agents hurt themselves trying to come out best. This simulation result is in line with the assumption of competitive preferences of some of the low cooperators in [20].

To conclude, our model of social preferences allows modeling agents that behave as if the actual participants possessed the same preferences. The agent-based simulation qualitatively reproduces the aggregate results of the cooperator classes in the experiment. Thus, our model meets the requirements of validation level 1 described by [41]. To explain the behavior of participants categorized as low cooperators, however, we need two classes of agents–rational and competitive low cooperators.

## 3.6 Endogenous punishment institutions

Following the simulation with fixed exogenous institutions, we extended the simulation by agents' voting decisions on the punishment institutions. Similar to [20], agents vote if they want to introduce punishment or not and if the punishment scheme should be mild or severe. The simple majority in a group decides. In contrast to the experimental design, agents give these two voting decisions simultaneously, i.e., we also know if they prefer mild or severe punishment when the majority of the group votes against punishment. Furthermore, the simulation starts with a 1500-period learning phase, in which the institutions are determined randomly every fourth round. This procedure ensures that agents "understand" every institution before their voting decisions every fourth of the next 100 rounds. This resembles the experimental instructions trying to assure subjects' understanding by examples and incentivized control questions. We use the same reinforcement-learning parameterization as in Section 3.5 (Table 4), except for reducing the attractivity depreciation parameter to $\varphi := 0.001$ and the probability to imitate to $p_{imit} := 0.05$. The first adaption guarantees successful learning in a

larger strategy space. The second is needed for our endogenous punishment institutions because our previous imitation mechanism is of limited use for changing institutions.

The agents' decisions on the institutions are based on the expected reward (more precisely utility) in each institution. We use the same reinforcement learning routine that is used to determine the agents' contribution. There are four different institution voting choices (no punishment/mild punishment if the majority voted for punishment; no/severe; yes/mild; yes/severe). The expected utility of the institutions is formed by:

$$q_{inst}(t) = \sum_{a \in A} p(a,t) \cdot q(a,t) \tag{9}$$

For each institution choice, the expected utility $q_{inst}(t)$ is formed by summing the product of probability $p(a,t)$, given that the institution is active, and the expected reward $q(a,t)$ for each possible contribution level. Following that, the probability an agent chooses an institution is given by:

$$p_{inst}(t) = \frac{exp^{\left(\frac{q_{inst}(t)}{q^{max} \cdot \mu(t)}\right)}}{\sum_{inst \in I} exp^{\left(\frac{q_{inst}(t)}{q^{max} \cdot \mu(t)}\right)}} \tag{10}$$

Agents do not learn from their voting decisions directly. Instead, they update the expected utility of their contributions after each period only for the active institution.

Table 6 presents the average results of the EconSim model with endogenous institutions in the last 50 periods after 50 simulation runs. Competitive low cooperators do not implement punishment and only earned around 100 tokens. Yet rational low cooperators chose severe punishment in 82% (18% no punishment) of the times and managed to get a payoff of 125, middle cooperators implement mild punishment (78%; 22% no punishment) and achieved 144 tokens, and high cooperators nearly achieved the highest possible payoff (of 150 without punishment) by choosing no punishment (80%; 20% mild punishment).

Thus, the agent types learned to implement the institutions that maximize their objective functions. This contradicts the institution implementation in [20]'s *Sorted* treatment, but corresponds to the *Sorted-Info* treatment (see Fig 5 in [20]). Whereas in *Sorted-Info*, over the 24 rounds, subjects learn to choose institutions that help them, this learning procedure is extremely limited in *Sorted*. Note that these two treatments only differ by the additional information of contributions in one round, subjects get this information either 24 or 25 times. The initial contribution level seems to be essential for the human learning process. Our agents' learning process is effective without this information. Yet, in total, they have the chance to learn in 1600 instead of 24 periods.

**Table 6. Contributions and payoffs under endogenous punishment in the last 50 periods of 50 simulation runs.**

| Agent type | Punishment institution | Average contribution | Average payoff |
|---|---|---|---|
| Rational low cooperator ($i = 1$) | Severe (82.00%) | 83.42 (36.72) | 125.10 (10.85) |
| High cooperator ($i = 2$) | No (80.00%) | 98.79 (3.17) | 148.40 (2.30) |
| Middle cooperator ($i = 3$) | Mild (78.00%) | 96.33 (13.79) | 144.01 (5.98) |
| Competitive low cooperator ($i = 4$) | No (100.00%) | 0.23 (0.68) | 100.11 (0.34) |

In brackets: percentage of the most chosen punishment institution respectively standard deviation.

## 3.7 Sensitivity analysis

To provide some insights about the influences of the parameters of the reinforcement-learning algorithm and to find a good parameter combination for the replication of the experiment, we made some preliminary simulation runs with five agents of type $i = 1$ (see Tables 2 and 3). This type of agent corresponds to the selfish homo oeconomicus. In line with our simulation studies in sections 3.6 and 3.7, we run the preliminary study with 300 periods each run and, to have some statistical validation, 100 simulation runs per parameter combination. Unlike in sections 3.5 and 3.6, punishment is deactivated. Thus, the game-theoretic prediction is zero contributions to the public good. We label the learning process in a single run as successfully converged when $\mu(t)$ (see Section 3.2) of each agent reaches the defined minimum threshold of 0.01 with maximum contributions of each agent of five for at least ten periods. S1 Table presents different parameter combinations and their respective performances. Parameter combinations 1–17 are conducted with a one-at-a-time approach to explore the effects of single parameter changes concerning the baseline combination 0, assuming not too much interaction between the parameters. Parameter combinations 18–20 are variations in multiple dimensions.

Imitation, namely copying another's action, seems to be a powerful mechanism leading to fast convergence (see S2 Fig). The agents quickly find the optimal or close-to-optimal contribution levels with high reliability. Combinations with a high probability of imitation are 12, 15–18, and 20. Especially, imitating the best agent leads to good results (13–15). As we pointed out in Section 3.3, copying the best agents' contribution may yield undesired outcomes in simulation runs with severe punishment. This is why we opted for the imitation of random agents in the last two sections. The weights in the exponential smoothing of the attractivity also seem to have a strong influence on convergence: Simulation runs perform better if the models focus more on new values. Lower attractivity depreciation also leads to better performance. Parameter combination 20 is the combination that we used in the previous sections because it performed in a fast and stable way.

To gain some understanding of the agent's utility function with additively connected social preferences, we calculate the marginal utility of the contribution of an agent with one of our five motives (see Table 3). Selfishness and competitiveness both have a strictly negative marginal utility, meaning that an agent with only these motives will show uncooperative behavior. In contrast, social welfare and altruism yield strictly positive marginal utilities. Agents with only these motives will be cooperative. The utility of an inequity-averse agent depends on the average contribution level of the other agents in the group: The marginal utility will be positive if the others' contribution level is higher than their own contribution, and vice versa.

Combining only the motives with marginal utilities independent of other agents' contributions, we can calculate the decisions of a homo oeconomicus characterized by this utility function. As we assume bounded rationality and have limited computing time, our agents might not find the optimum in every case, especially when the marginal utility of the contribution approaches zero. Moreover, including inequity aversion makes it difficult to tell whether an agent will contribute at low or high levels. This is why we provide some simulation runs with varying weights of inequity aversion ($u_{i2}$) and constant weights for the other motives ($u_{i1} = 0.7$, $u_{i3} = 0$, $u_{i4} = 0$, $u_{i5} = 0.3$). We calculate a slight tendency to uncooperative behavior with a marginal utility of $\frac{\partial U_i}{\partial g_i} = -0.03$. However, our simulation results show medium contribution levels. With an increasing weight of inequity aversion, the resulting contributions at the end of our simulation runs are close to the game theoretic prediction (Fig 6). In similar simulation runs with a more cooperative tendency ($u_{i1} = 0.66$, $u_{i3} = 0$, $u_{i4} = 0$, $u_{i5} = 0.34$; marginal utility $\frac{\partial U_i}{\partial g_i} = 0.03$), we observe high contribution levels that decrease with higher weights of inequity aversion (Fig 7). For both parameter constellations, cooperation declines over time. This raises

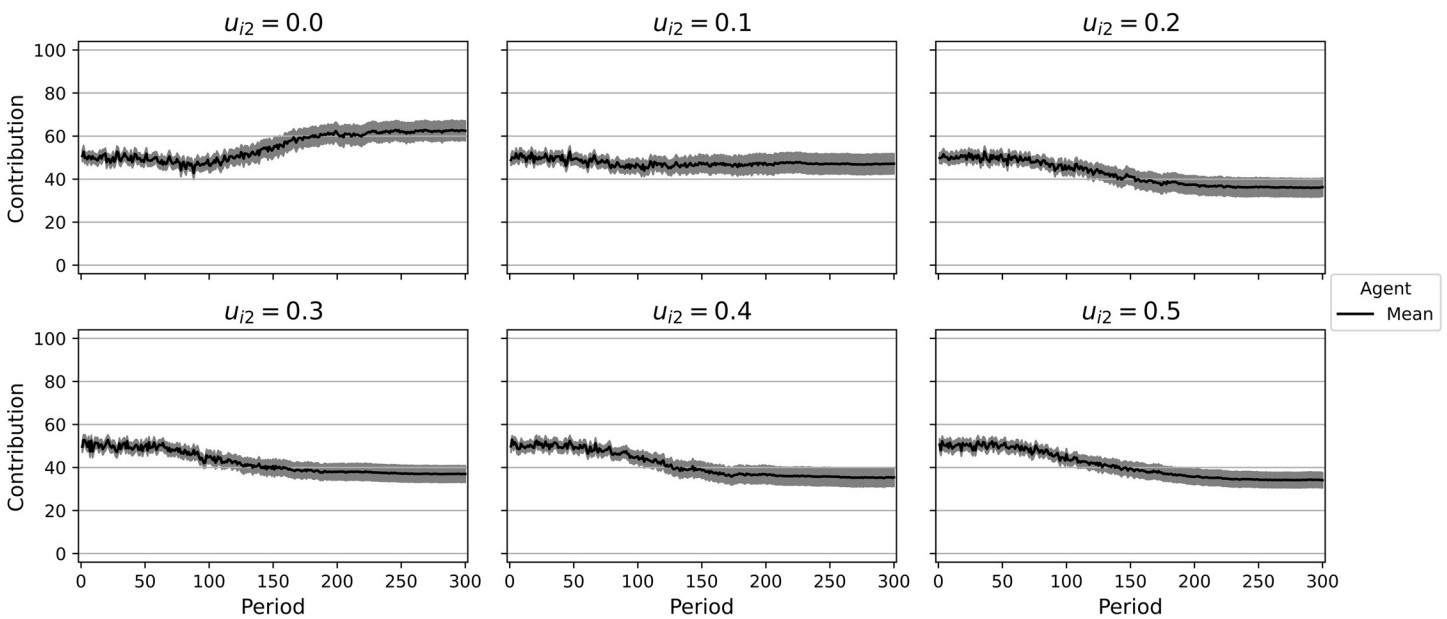

**Fig 6. Contributions of rather selfish agents with different weights of inequity aversion.** Selfishness: 0.70, competitiveness: 0, social welfare: 0, altruism: 0.30. Marginal utility of the own contribution: -0.03. In gray: confidence interval (95%) around the mean.

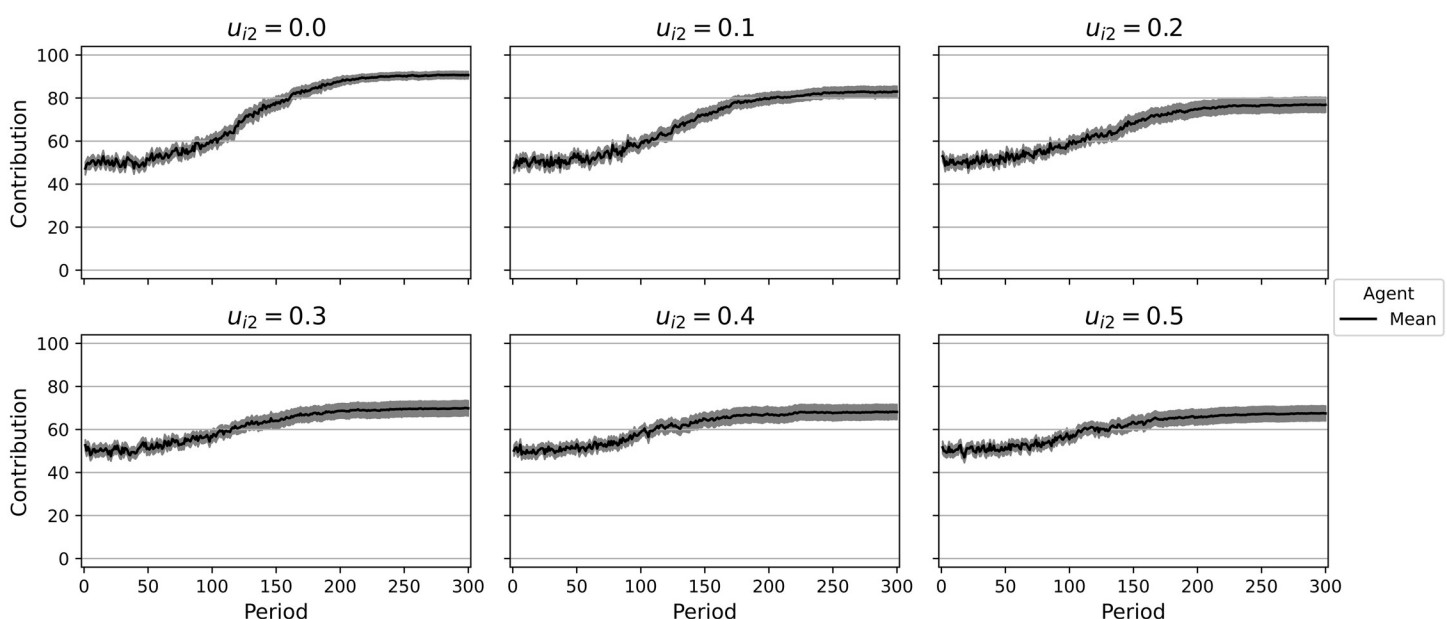

**Fig 7. Contributions of rather cooperative agents with different weights of inequity aversion.** Selfishness: 0.66, competitiveness: 0, social welfare: 0, altruism: 0.34. Marginal utility of the own contribution: 0.03. In gray: confidence interval (95%) around the mean.

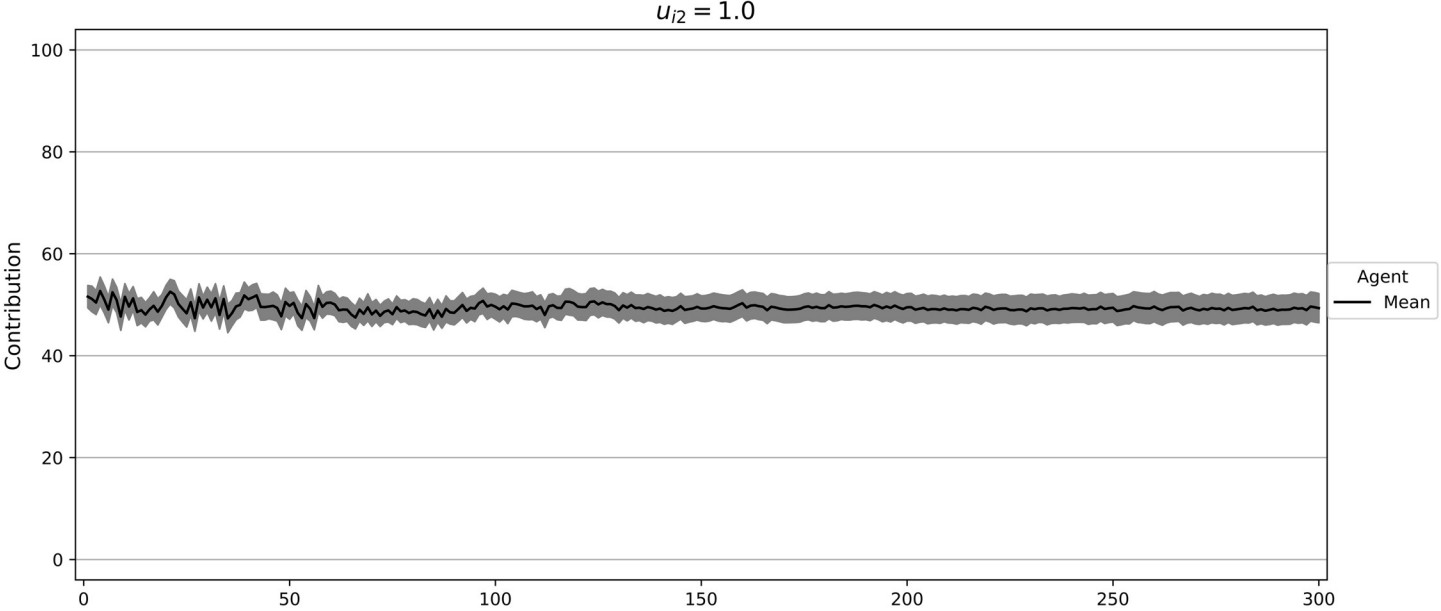

**Fig 8. Contributions of purely inequity-averse agents.** Selfishness: 0, inequity aversion: 1 competitiveness: 0, social welfare: 0, altruism: 0. Marginal utility of the own contribution: 0. In gray: confidence interval (95%) around the mean.

the question of whether inequity aversion yields a tendency for defection in our model. However, the simulation runs with a purely inequity-averse agent ($u_{i2} = 1$) suggest that there is not such a tendency (Fig 8).

When we introduce punishment, the game theoretic properties change only for the selfish motive because it is the only motive that depends on the payoff. Punishment adds a discontinuity into the function $u_{i1}$ and decreases the partial utility of the selfishness motive by the penalty level $p(k)$ in case of $g_i < 100$. A rational agent would ignore the punishment if its utility level is higher for any contribution level lower than 100, or formally $\exists g' \in \{0, 1, . . ., 99\}$ with $U_i(g_i = g') > U_i(g_i = 100)$. As we weigh the motives to get a combined utility function $U_i$, we can calculate if punishment is high enough to affect the agent at a specific contribution level $g_i = (100—h)$ with $h \in \{1, 2, . . ., 100\}$. Rational agents will cooperate when there is no h for which the following inequality is true:

$$w_{i1} \cdot \frac{p(k)}{1.5} < \int_{100-h}^{100^-} \frac{\partial U_i}{\partial g_i}(h')dh' \tag{11}$$

Thus, we can predict whether agents will be influenced by punishment or not. As we assume homogenous agents, the game-theoretic properties will not change with varying weights of inequity aversion because every agent will prefer the same level of contribution. It is sufficient to analyze the utility at the contribution level $g_i = 0$. In our simulation model with boundedly rational agents, the game-theoretic prediction seems to attract behavior. Still, this depends on the strength of attraction (marginal utility of contributions) and the dynamics of learning.

## 4. Conclusion and discussion

We introduce an agent-based model of a public goods game with punishment institutions, in which we build agents based on a social preference model inspired by previous theory and experimental evidence. We are able to calibrate our model of social preferences in a way that agents in the simulation show very similar behavior to humans in the experiment of [20]. The reinforcement-learning algorithm can replicate human behavior in our model with the assumption of simple motives. Coming back to the petri dish analogy of the introduction, we accomplished the replication of Petri dishes similar to the original Petri dish, the experimental result. Therefore, our reinforcement-learning algorithm can be used to imitate human behavior in more complex settings in the future. This might include sophisticated agent-based models with complex trading and production of goods. Furthermore, the created model of the public goods game can be used to test how well it can predict human decisions in different designs.

Additionally, we introduced a new model of social preferences, which combines different motives by an additive connection. We allow contradictive motives because they may be part of everyday decision-making. Nevertheless, we still cannot be sure if participants in the experiment actually behave according to these motives. For instance, we do not consider trust and reciprocity in our model. Our model of social preferences is only outcome-based and not intention-based. While this protects our agent-based model against self-fulfilling beliefs [42], intention-based preferences may be more realistic [5] and a promising venture for future research in agent-based simulations. Including an intention-based model could help to address the question of how agents react to information about other agents. Our model is limited in this respect.

Another potential for future research is to endogenize the social preferences of agents. The standard assumption in behavioral economics is that social preferences are given traits of subjects that do not change. Yet it might be more realistic to assume that social preferences are characteristics that are influenced by treatments or the behavior of other subjects. In an agent-based model, a start population with given social preferences may develop into a generation with another structure of social preferences. To this end, reproduction and mutation rules should consider the efficiency of different forms of social preferences in different institutions. Finally, we see potential in implementing our motives into other settings of agent-based models, replications of experiments, or complex economic simulations of society. For example, they could be able to explain preferences for the redistribution of income. Furthermore, our model of social preferences can be used in a simulation model that studies the effects of inequality on voting behavior.

## Supporting information

**S1 Fig. Graph of the agent-based model in EconSim-GUI.**
(TIF)

**S2 Fig. Boxplots of the convergence time of different parameter combinations (see S1 Table for detailed parameter settings).** In red: chosen parameter configuration in sections 3.6 and (slightly adjusted) 3.7.
(TIF)

**S3 Fig. Dynamics of contributions and payoffs of low cooperators under no, mild, and severe punishment in a single run including the arithmetic mean of the agents.**
(TIF)

**S4 Fig. Dynamics of contributions and payoffs of high cooperators under no, mild, and severe punishment in a single run including the arithmetic mean of the agents.**
(TIF)

**S5 Fig. Dynamics of contributions and payoffs of middle cooperators under no, mild, and severe punishment in a single run including the arithmetic mean of the agents.**
(TIF)

**S6 Fig. Dynamics of contributions and payoffs of competitive low cooperators under no, mild, and severe punishment in a single run including the arithmetic mean of the agents.**
(TIF)

**S1 Table. Sensitivity analysis of the reinforcement learning parameters.** Underlined: changes relative to Combination 0.
(PDF)

**S1 File. This file contains a pseudocode of the used reinforcement learning algorithm (NESASS) and some additional comments.**
(PDF)

## Author Contributions

**Conceptualization:** Christoph Bühren, Janis Kesten-Kühne.

**Data curation:** Jan Haarde, Christian Hirschmann, Janis Kesten-Kühne.

**Formal analysis:** Christoph Bühren, Christian Hirschmann, Janis Kesten-Kühne.

**Investigation:** Christoph Bühren, Christian Hirschmann, Janis Kesten-Kühne.

**Methodology:** Christoph Bühren, Christian Hirschmann, Janis Kesten-Kühne.

**Project administration:** Christoph Bühren.

**Software:** Jan Haarde, Christian Hirschmann, Janis Kesten-Kühne.

**Supervision:** Christoph Bühren, Janis Kesten-Kühne.

**Validation:** Christoph Bühren, Jan Haarde, Christian Hirschmann, Janis Kesten-Kühne.

**Visualization:** Christoph Bühren, Jan Haarde, Christian Hirschmann, Janis Kesten-Kühne.

**Writing – original draft:** Christoph Bühren, Christian Hirschmann.

**Writing – review & editing:** Christoph Bühren, Jan Haarde, Christian Hirschmann, Janis Kesten-Kühne.

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
