## [Decision Letter · Decision Letter 0]

5 Dec 2022

PONE-D-22-30373Social Preferences in the Public Goods Game – An Agent-Based Simulation with EconSimPLOS ONE

Dear Dr. Christoph Bühren,

Thank you for submitting your manuscript to PLOS ONE. After careful consideration, we feel that it has merit but does not fully meet PLOS ONE’s publication criteria as it currently stands. Therefore, we invite you to submit a revised version of the manuscript that addresses the points raised during the review process.

We received three reports on your manuscript, and all of them see potentiality in your work. However, they recommend improving the paper through a major revision.

We look forward to receiving your revised manuscript.

Kind regards,

Jaume Garcia-Segarra

Academic Editor

PLOS ONE

Reviewers' comments:

Reviewer's Responses to Questions

**Comments to the Author**

1. Is the manuscript technically sound, and do the data support the conclusions?

Reviewer #1: Yes

Reviewer #2: Yes

Reviewer #3: Partly

2. Has the statistical analysis been performed appropriately and rigorously? 

Reviewer #1: Yes

Reviewer #2: Yes

Reviewer #3: No

3. Have the authors made all data underlying the findings in their manuscript fully available?

Reviewer #1: Yes

Reviewer #2: No

Reviewer #3: No

4. Is the manuscript presented in an intelligible fashion and written in standard English?

Reviewer #1: Yes

Reviewer #2: Yes

Reviewer #3: Yes

5. Review Comments to the Author

Reviewer #1: The authors focus their study on the problem of social preferences in the public goods by introducing a reinforcement learning approach where they model an agent based simulation of a public goods game with endogenous punishments institutions. Additionally, the authors introduce an outcome based model of social preferences that is capable to determine the agents utility, the contribution, as well as the voting behavior during the learning process. The manuscript is overall well written and easy to follow and the authors have well thought out their main contributions. The provided theoretical analysis is concrete, complete, and correct and the authors have provided all the intermediate steps in order to enable the average reader to easily follow it. Furthermore, the authors have provided a very detailed set of numerical results in order to show the pure operation and the performance of the proposed framework. The authors are highly encouraged to consider the following suggestions provided by the reviewer in order to improve the scientific depth of their manuscript, as well as they need to address the following comments in order to improve the quality of presentation of their manuscript. Initially, the provided related work in section one and two, needs to be substantially revised in order to be presented by using more summative language in order to better identify the research contributions that have already been performed in the literature and the research gap that the authors try to address. In Section 1, the authors need to discuss existing approaches that have been introduced in the recent literature, such as A Paradigm Shift Toward Satisfaction, Realism and Efficiency in Wireless Networks Resource Sharing, doi: 10.1109/MNET.011.2000368 that they consider the concepts of satisfaction, humans behavior in limited information environments, and they discussed the concept of satisfaction games in order to overcome those burdens. The provided related work needs to be substantially revised in order to capture the most recent advances in the field. Furthermore, the authors need to include an additional section in their manuscript describing the implementation cost of the proposed study for the readers that are interested in reproducing the proposed model. Finally, the overall manuscript needs to be checked for typos, syntax, and grammar errors in order to improve the quality of its presentation.

Reviewer #2: In this manuscript, the authors apply the reinforcement learning algorithm to model an agent-based simulation of the public goods game with endogenous punishment, as well as develop an outcome-based model of social preferences to determine the agent’s utility, contribution, and voting behavior during the learning procedure. It is found that the proposed model can replicate human behavior and has provided an important insight into understanding the underlying motives of human behavior. Besides, the proposed method can be extended to more complicated simulations. After reading the paper, I think that it is an interesting work and have some following comments.

1. It is interesting to incorporate the reinforcement learning into evolutionary game. But I think that it is necessary to clarify why agents in evolutionary games can use the learning algorithm to update their actions. In order to help readers understand this point, the authors can given some description.

2. In the subsection 3.5 and 3.6, two different types of punishment institutions are mentioned, that is, exogenous punishment institutions and endogenous punishment institutions. These two concepts of punishment institutions may be needed to be explained. Besides, it would be better to cite some related works regarding the exogenous (endogenous) punishment. For example, Journal of the Royal Society Interface, 12 (2015) 20140935; Physics Letters A, 386(2021) 126965; Europhysics Letters, 136 (2022) 68002.

3. Page 3: In the first paragraph of subsection 2.1, the description of the first equation (i.e., \\pi=…) could come first, and then write the expression of equation. For example, “In the sorting game, subjects were randomly shuffled into groups of five to play a standard one-shot public goods game…0.3.” should be changed to “In the sorting game, subjects were randomly shuffled into groups of five to play a standard one-shot public goods game. In the game, the endowment was set to 350 tokens, and the participants simultaneously decided on their individual contribution $_i $ to the public good. The marginal product of the public good is $=1.5$ and the marginal per capita return $/= 0.3$. In this case, the payoff function can be written as $pi=…$”. In addition, the digital labels should be added for the equations.

4. There exist some typos, please correct them. For example, in page 3: in the first paragraph of subsection 3.2, “public good” should be changed to “public goods”.

Reviewer #3: The paper provides a simple agent-based model based on reinforcement learning and social preferences to reproduce the results of previous laboratory experiments. In my opinion, this is a nice idea, and the paper seems to be interesting and successful in reproducing experimental findings through computer simulations. However, I have some comments aimed at clarifying some of the more critical points I detected after reading the paper.

Section 2. The authors introduce a previous paper on a public goods with endogenous punishment with a sorting mechanism based on participants’ previous contributions in a one-shot game. Then, they discuss how groups of agents learn something during the game (just before Subsection 2.1). It is not clear to me if we are still referring to a one-shot game or a repeated one.

In Subsection 2.1, a payoff function with an endowment of 350 token is commented. One may wonder if the number 350 has a particular validity within this framework. Moreover, why the choice for the marginal product of public good was a=1.5 and not another value.

I am a bit confused about the difference between Sorted-Info and Sorted treatments. Please provide a clearer explanation in the text.

A second payoff function is then introduced, this time with 100. Why now the number is no longer 350 but rather 100? And why that specific value?

The next payoff function includes also a punishment technology. Within this setting, the authors assume that: “Every person contributing less than the maximum of 100 tokens to the public good automatically had to pay a fine of 50 under mild punishment and 90 in the severe punishment scheme.” My impression is that such an assumption is a quite strong one because it implies that contributing 99 or 1 has the same consequences for punishment. It does not make a lot of sense in my view (but under very extreme scenarios).

It would be nice to provide an explanation of what EcoSim is when it is firstly introduced in the text.

In Subsection 3.1, the authors provide a short explanation of the simulation software they used. Anyway, I do not see a specific reason why this kind of simulation software could be considered an effective one. Given the simplicity of the setting, this is something that one would do from the scratch or with many other programming tools. Thus, I would suggest to downsize the role of the simulation software because, at least this was my understanding, there is not a specific motivation for using this particular tool or its modular architecture.

Beginning of page 7. I do not see the fundamental difference between r and q. Why using q in relation to a(t-1) and r in relation to a(t)?

Last equation of page 7. The equation is not for defining the interval between the minimum and maximum temperature as maintained by the authors. Rather, it is about the evolution of the temperature within a given interval. Moreover, it is not so clear the comment about when the update scheme is active or not.

In my view, Section 3.4 is too much long and contains comments that can be condensed in less lines. Mostly important, the results should be presented in a more organized and schematic way.

I did not understand why one should normalize utilities to a range from 0 to 100 when higher values can occur and, indeed, they are commented in subsequent lines and sections.

Page 12. “This is because the Nash Equilibrium does not change with mild punishment.” I guess this is not a general result as it may depend on the definition of a ‘mild’ punishment institution (basically the values of fees and sanctions).

Table 3. It would be interesting to understand why a specific choice of weights is proposed to characterize a type of agent.

Table 4. Many parameters feature the learning algorithm. The choice of specific values should be motivated, based on the literature or perhaps the empirical and/or experimental evidence.

In general, the authors should perform some sensitivity analyses on single parameters as well as on their combinations in order to understand their role in shaping simulation results. The graphical analysis proposed in Figures at pages 24 and 25 is quite rough and could be refined, also including some statistics on the simulation results. Moreover, the paper would gain in readability from a better organization and a more schematic presentations of results.

6. PLOS authors have the option to publish the peer review history of their article (what does this mean?). If published, this will include your full peer review and any attached files.

Reviewer #1: No

Reviewer #2: No

Reviewer #3: No

---

## [Author Response · Author response to Decision Letter 0]

19 Jan 2023

Reviewer #1: 

"The authors focus their study on the problem of social preferences in the public goods by introducing a reinforcement learning approach where they model an agent based simulation of a public goods game with endogenous punishments institutions. Additionally, the authors introduce an outcome based model of social preferences that is capable to determine the agents utility, the contribution, as well as the voting behavior during the learning process. The manuscript is overall well written and easy to follow and the authors have well thought out their main contributions. The provided theoretical analysis is concrete, complete, and correct and the authors have provided all the intermediate steps in order to enable the average reader to easily follow it. Furthermore, the authors have provided a very detailed set of numerical results in order to show the pure operation and the performance of the proposed framework. The authors are highly encouraged to consider the following suggestions provided by the reviewer in order to improve the scientific depth of their manuscript, as well as they need to address the following comments in order to improve the quality of presentation of their manuscript."

Thank you. 

"Initially, the provided related work in section one and two, needs to be substantially revised in order to be presented by using more summative language in order to better identify the research contributions that have already been performed in the literature and the research gap that the authors try to address."

We revised these sections and tried to describe our contributions more clearly.

"In Section 1, the authors need to discuss existing approaches that have been introduced in the recent literature, such as A Paradigm Shift Toward Satisfaction, Realism and Efficiency in Wireless Networks Resource Sharing, doi: 10.1109/MNET.011.2000368 that they consider the concepts of satisfaction, humans behavior in limited information environments, and they discussed the concept of satisfaction games in order to overcome those burdens."

We cited this paper.

"The provided related work needs to be substantially revised in order to capture the most recent advances in the field." 

Furthermore, we added more relevant and recent related literature.

"Furthermore, the authors need to include an additional section in their manuscript describing the implementation cost of the proposed study for the readers that are interested in reproducing the proposed model."

We now provide the pseudocode and additional comments on the agent’s decision-making in the Appendix. Moreover, we added a program flow chart in our manuscript showing the essential procedure of our simulation model, and we revised the description of EconSim.

"Finally, the overall manuscript needs to be checked for typos, syntax, and grammar errors in order to improve the quality of its presentation."

We revised the whole manuscript, and a professional copy editor helped us to improve the quality of the text. Thank you for your thoroughly reading of our manuscript and your very helpful suggestions.

Reviewer #2: 

"In this manuscript, the authors apply the reinforcement learning algorithm to model an agent-based simulation of the public goods game with endogenous punishment, as well as develop an outcome-based model of social preferences to determine the agent’s utility, contribution, and voting behavior during the learning procedure. It is found that the proposed model can replicate human behavior and has provided an important insight into understanding the underlying motives of human behavior. Besides, the proposed method can be extended to more complicated simulations. After reading the paper, I think that it is an interesting work and have some following comments."

Thank you.

"1. It is interesting to incorporate the reinforcement learning into evolutionary game. But I think that it is necessary to clarify why agents in evolutionary games can use the learning algorithm to update their actions. In order to help readers understand this point, the authors can given some description."

We clarified why reinforcement learning can be used for decision-making in agent-based models.

"2. In the subsection 3.5 and 3.6, two different types of punishment institutions are mentioned, that is, exogenous punishment institutions and endogenous punishment institutions. These two concepts of punishment institutions may be needed to be explained."

We now explain these punishment institutions in more detail.

"Besides, it would be better to cite some related works regarding the exogenous (endogenous) punishment. For example, Journal of the Royal Society Interface, 12 (2015) 20140935; Physics Letters A, 386(2021) 126965; Europhysics Letters, 136 (2022) 68002."

Furthermore, we cited related literature on exogenous and endogenous punishment.

"3. Page 3: In the first paragraph of subsection 2.1, the description of the first equation (i.e., \\pi=…) could come first, and then write the expression of equation. For example, “In the sorting game, subjects were randomly shuffled into groups of five to play a standard one-shot public goods game…0.3.” should be changed to “In the sorting game, subjects were randomly shuffled into groups of five to play a standard one-shot public goods game. In the game, the endowment was set to 350 tokens, and the participants simultaneously decided on their individual contribution $_i $ to the public good. The marginal product of the public good is $=1.5$ and the marginal per capita return $/= 0.3$. In this case, the payoff function can be written as $pi=…$”. In addition, the digital labels should be added for the equations."

We adopted these suggestions in Section 2.1, thank you. Furthermore, we numbered the equations.

"4. There exist some typos, please correct them. For example, in page 3: in the first paragraph of subsection 3.2, “public good” should be changed to “public goods”."

We corrected this. Thank you for your thoroughly reading of our manuscript and your very helpful suggestions. 

Reviewer #3: 

"The paper provides a simple agent-based model based on reinforcement learning and social preferences to reproduce the results of previous laboratory experiments. In my opinion, this is a nice idea, and the paper seems to be interesting and successful in reproducing experimental findings through computer simulations. However, I have some comments aimed at clarifying some of the more critical points I detected after reading the paper."

Thank you.

"Section 2. The authors introduce a previous paper on a public goods with endogenous punishment with a sorting mechanism based on participants’ previous contributions in a one-shot game. Then, they discuss how groups of agents learn something during the game (just before Subsection 2.1). It is not clear to me if we are still referring to a one-shot game or a repeated one."

We now distinguish more clearly between the “sorting game” and the “institution formation game” of Bühren and Dannenberg (2021).

In Subsection 2.1, a payoff function with an endowment of 350 token is commented. One may wonder if the number 350 has a particular validity within this framework. Moreover, why the choice for the marginal product of public good was a=1.5 and not another value. 

In the revised manuscript, we explain in more detail the parameter choices of the experiment, which we replicated with our simulation.

"I am a bit confused about the difference between Sorted-Info and Sorted treatments. Please provide a clearer explanation in the text."

Moreover, we describe the difference between their Sorted and Sorted-Info treatments much clearer.

"A second payoff function is then introduced, this time with 100. Why now the number is no longer 350 but rather 100? And why that specific value?"

Bühren and Dannenberg chose 350 for the sorting game to increase the likelihood of enough variance of participants’ contributions compared to the more standard endowment of 100, in which participants’ initial contributions are often clustered around 50. In the institution formation game, the authors stuck to the more standard endowment to make their results better comparable to previous literature. In the revised manuscript, we discuss these parameter choices. As we replicated their experiment, we chose the same parameters.

"The next payoff function includes also a punishment technology. Within this setting, the authors assume that: “Every person contributing less than the maximum of 100 tokens to the public good automatically had to pay a fine of 50 under mild punishment and 90 in the severe punishment scheme.” My impression is that such an assumption is a quite strong one because it implies that contributing 99 or 1 has the same consequences for punishment. It does not make a lot of sense in my view (but under very extreme scenarios)."

Bühren and Dannenberg implemented extreme punishment institutions similar to Tyran and Feld (2006). The reason is that they provided institutions in their experiment that are able to increase cooperation (and payoffs) considerably. However, the authors also implemented large implementation costs for these institutions: 5 tokens per round for mild and 20 tokens per round for severe punishment. We discuss these parameters in more detail in the revised manuscript.

"It would be nice to provide an explanation of what EcoSim is when it is firstly introduced in the text."

We now provide a brief explanation of EconSim when it is firstly introduced.

"In Subsection 3.1, the authors provide a short explanation of the simulation software they used. Anyway, I do not see a specific reason why this kind of simulation software could be considered an effective one. Given the simplicity of the setting, this is something that one would do from the scratch or with many other programming tools. Thus, I would suggest to downsize the role of the simulation software because, at least this was my understanding, there is not a specific motivation for using this particular tool or its modular architecture."

In the revised Section 3.1, we downsize the role of EconSim and discuss why we chose this program.

"Beginning of page 7. I do not see the fundamental difference between r and q. Why using q in relation to a(t-1) and r in relation to a(t)?"

We added some comments to clarify the difference of the reward and attractivity vector.

"Last equation of page 7. The equation is not for defining the interval between the minimum and maximum temperature as maintained by the authors. Rather, it is about the evolution of the temperature within a given interval. Moreover, it is not so clear the comment about when the update scheme is active or not."

Thank you for this note. We corrected and clarified it.

"In my view, Section 3.4 is too much long and contains comments that can be condensed in less lines."

We shortened Section 3.4 substantially.

"Mostly important, the results should be presented in a more organized and schematic way."

We revised the results section. We added new tables and figures, and we included a subsection with sensitivity analyses (the new Section 3.7.). Furthermore, we added subheadings in Section 3.5. 

"I did not understand why one should normalize utilities to a range from 0 to 100 when higher values can occur and, indeed, they are commented in subsequent lines and sections."

We normalized the utilities to make them better comparable across motives and types of agents, which we explain in the revised manuscript. 

"Page 12. “This is because the Nash Equilibrium does not change with mild punishment.” I guess this is not a general result as it may depend on the definition of a ‘mild’ punishment institution (basically the values of fees and sanctions)."

You are right. We clarified this statement: In our case, mild punishment is designed as a non-deterrent punishment scheme similar to Tyran and Feld (2006). 

"Table 3. It would be interesting to understand why a specific choice of weights is proposed to characterize a type of agent."

We added a discussion of why we chose the specific weights in Table 3. As this choice might seem arbitrary, we added a subsection with sensitivity analyses (Section 3.7), in which we discuss the determinants for the tendency toward cooperative behavior of our agents.

"Table 4. Many parameters feature the learning algorithm. The choice of specific values should be motivated, based on the literature or perhaps the empirical and/or experimental evidence".

We clarified the parameter choices of Table 4 and compare different parameter combinations and their performance in 100 simulation runs in our sensitivity analyses.

"In general, the authors should perform some sensitivity analyses on single parameters as well as on their combinations in order to understand their role in shaping simulation results."

We provide a sensitivity analysis of the learning parameters and a mathematical analysis of the utility function and its components. Table A1 in the Appendix shows 20 different parameter constellations that we tried out, and Figure A2 presents their convergence time. We adopted a new parameter combination that performed more stable and better than our first one. Thank you for supposing the sensitivity analysis. This analysis enhanced our simulation model and its performance.

"The graphical analysis proposed in Figures at pages 24 and 25 is quite rough and could be refined, also including some statistics on the simulation results."

We revised these figures. We added a cooldown mechanism on the parameter epsilon-greedy to prevent explorative behavior, which caused confusing peaks in the old figures. To prevent showing stochastic anomalies, we added mean results of 100 simulation runs and show confidence intervals in the figures. In the revised manuscript, we present these simulation results (for the exogenous punishment institutions) in figures 2-5 (and sensitivity analyses in figures 6-8). Furthermore, we show the corresponding figures of single runs in the Appendix (figures A3-A6). Moreover, we present more statistics on these simulation results in Table 5 (and for the endogenous punishment institutions in Table 6).

"Moreover, the paper would gain in readability from a better organization and a more schematic presentations of results."

We copy-edited the whole manuscript, and we tried to present our results in a more organized and schematic way. Thank you for your thoroughly reading of our manuscript and your very helpful suggestions.

---

## [Decision Letter · Decision Letter 1]

8 Feb 2023

Social Preferences in the Public Goods Game – An Agent-Based Simulation with EconSim

PONE-D-22-30373R1

Dear Dr. Bühren,

We’re pleased to inform you that your manuscript has been judged scientifically suitable for publication and will be formally accepted for publication once it meets all outstanding technical requirements.

Kind regards,

Jaume Garcia-Segarra

Academic Editor

PLOS ONE

Additional Editor Comments (optional):

Reviewers' comments:

Reviewer's Responses to Questions

**Comments to the Author**

1. If the authors have adequately addressed your comments raised in a previous round of review and you feel that this manuscript is now acceptable for publication, you may indicate that here to bypass the “Comments to the Author” section, enter your conflict of interest statement in the “Confidential to Editor” section, and submit your "Accept" recommendation.

Reviewer #1: All comments have been addressed

Reviewer #2: All comments have been addressed

Reviewer #3: All comments have been addressed

2. Is the manuscript technically sound, and do the data support the conclusions?

Reviewer #1: Yes

Reviewer #2: Yes

Reviewer #3: Yes

3. Has the statistical analysis been performed appropriately and rigorously? 

Reviewer #1: Yes

Reviewer #2: Yes

Reviewer #3: Yes

4. Have the authors made all data underlying the findings in their manuscript fully available?

Reviewer #1: Yes

Reviewer #2: Yes

Reviewer #3: Yes

5. Is the manuscript presented in an intelligible fashion and written in standard English?

Reviewer #1: Yes

Reviewer #2: Yes

Reviewer #3: Yes

6. Review Comments to the Author

Reviewer #1: The authors have addressed in detail the reviewers' comments. A very detailed revision of the paper has been performed and detailed answers to the reviewers' comments have been provided. This reviewer has no further concerns about this paper.

Reviewer #2: The authors have addressed my comments accordingly and I would like to recommend the publication of the work in PLoS ONE.

Reviewer #3: I am satisfied with the revision of the paper in response to the points I raised during the first round of the reviewing process.

7. PLOS authors have the option to publish the peer review history of their article (what does this mean?). If published, this will include your full peer review and any attached files.

Reviewer #1: No

Reviewer #2: No

Reviewer #3: No

---

## [Editor Report · Acceptance letter]

16 Feb 2023

PONE-D-22-30373R1 

Social Preferences in the Public Goods Game – An Agent-Based Simulation with EconSim 

Dear Dr. Bühren:

I'm pleased to inform you that your manuscript has been deemed suitable for publication in PLOS ONE. Congratulations! Your manuscript is now with our production department. 

Kind regards, 

on behalf of

Dr. Jaume Garcia-Segarra 

Academic Editor

PLOS ONE